# A Comparison of Strategies to Improve Uptake of COVID-19 Vaccine among High-Risk Adults in Nairobi, Kenya in 2022

**DOI:** 10.3390/vaccines11020209

**Published:** 2023-01-17

**Authors:** Joan Yego, Robert Korom, Emma Eriksson, Sharon Njavika, Oulimata Sane, Purity Kanorio, Oliver Rotich, Stellah Wambui, Eunice Mureithi

**Affiliations:** 1Penda Health Ltd., Nairobi P.O. Box 22647-00100, Kenya; 2Dalberg Advisors, Nairobi P.O. Box 100657-00100, Kenya

**Keywords:** COVID-19, high-risk groups, vaccine uptake interventions, Kenya

## Abstract

Background: COVID-19 vaccine uptake in Kenya is still low compared to other countries, especially in Europe and North America. In most parts of the country, a large percentage of the Kenyan population remains unvaccinated. As of October 2022, the Ministry of Health (Kenya) estimates that only 36.2% of the adult population had been fully vaccinated. Methods: We conducted an experimental study in April 2022 targeting unvaccinated adults who had a history of hypertension and/or diabetes and those in the 60+ age group. We tested various messaging approaches using two different intervention channels. Results: Although the overall rate of vaccinated individuals according to national records is low, responses from the study group collected through phone call conversations show that higher-risk adults such as those older than 60 or those with chronic illnesses have a remarkably high vaccination rate of 89%. After the study, four participants received a COVID-19 vaccine within 1 month of the intervention. These four participants all received a loss-messaging intervention approach during the study. Conclusion: This study supports a national approach to increasing COVID-19 vaccination rates using loss-messaging directed at unvaccinated, high-risk individuals.

## 1. Introduction

As COVID-19 continues to affect the lives of many people around the world, the effects of the disease are much more severe among high-risk populations. According to the Center for Disease Control and Prevention (CDC), high-risk groups include older adults in the 50+ age group as well as people with existing medical conditions such as cancer, HIV, diabetes, heart disease and obesity [1]. Among adults with existing chronic illness, the effects of COVID-19 not only impact on the physical health of the individuals but also their mental health [2]. Studies also show that the mortality rate from COVID-19 infection is increased among high-risk groups [3,4].

Many communities in different regions have embraced the COVID-19 vaccine to protect themselves and their families. As of October 2022, over 12.7 billion doses of the vaccine had been administered globally [5]. There is extensive and overwhelming evidence for the safety and effectiveness of vaccines against disease infection, and this evidence also applies to the various COVID-19 vaccine options [6,7]. Studies have shown that vaccination against COVID-19 has contributed to a reduced number of new COVID infections as well as reduced morbidity and mortality if one is infected with the virus [8]. 

Despite the evidence of vaccine efficacy and safety, COVID-19 vaccine uptake is still a major challenge in many parts of the world. When comparing vaccination rates based on social-economic status, high and middle-income countries have higher rates of vaccination with 82% of their population having received at least one dose of the vaccine compared to only 24% of the population in low-income countries [9]. There are many factors that contribute to this disparity in vaccination rates by region including availability of approved vaccines, efficacy of primary health care systems, and varying levels of vaccine hesitancy [10,11]. We recently published in this journal the first study of perceptions and knowledge about the COVID-19 vaccines in a cross section of the Kenyan population. That study demonstrated that 80% of Kenyans surveyed felt it was important to receive a vaccine to protect others from COVID-19; however, 40% still reported vaccine hesitancy related to side effects of the vaccine, suggesting that increasing vaccine uptake remains a dynamic challenge in Kenya [12]. As vaccine hesitancy has been a challenge for public health practitioners for decades, the COVID-19 pandemic and other global issues including food insecurity, climate change, and political instability have further affected people’s habits and health practices. These global challenges have particularly affected the habits of young adults [13].

In Kenya, periodic updates provided by the Ministry of Health in 2022 show that COVID-19 vaccines have been readily available especially in urban areas, since mid-2021 [14]. Despite the availability of the vaccines a large proportion of the population remains unvaccinated. As of October 2022, the Ministry of Health (Kenya) estimated that only 36% of the adult population had been fully vaccinated [14]. This relatively low uptake of the COVID-19 vaccine means that millions of Kenyans are at a high risk of not only contracting the virus but also experiencing health complications as a result of the infection. It is important to acknowledge that low uptake of the COVID-19 vaccine in our setting is a result of numerous structural inequities and social determinants of health, which include poverty, distrust in the healthcare system, health literacy, and others. High-risk patients in our setting are particularly vulnerable and deserve additional health system resources and efforts to offer adequate protection from serious illness.

The urgency of mass vaccination campaigns brought about by COVID-19 led to renewed interest in strategies to counter vaccine misinformation and to mitigate vaccine hesitancy. Paltiel and colleagues found that implementation of a mass vaccination program and countering vaccine hesitancy was in fact even more important than clinical efficacy of the vaccine as demonstrated in clinical trials to achieve optimal clinical outcomes at the population level [15]. A review of organizational and individual-level strategies to mitigate vaccine hesitancy suggested that a trusted healthcare provider making a strong recommendation to receive the vaccine was a key element of increasing patient uptake of the vaccines [16]. Furthermore, the review suggested that “gain messaging”—communication that focuses on positive upsides for patients—is particularly effective in a North American context.

A recent study from Limaye and colleagues assessed three different messaging approaches (health outcomes, economic impacts, and social norming) given by one of two messengers (a healthcare worker or a peer) [17]. This study demonstrated that in the Kenyan context, a discussion of health outcomes, specifically the risk of negative health outcomes from COVID-19 disease without having received the vaccine, was considerably more effective compared to messaging approaches that focused on social norming or economic impacts on the individual. The vast majority (more than 90%) of participants surveyed in that study were under age 40, and thus at a low baseline risk of complications from COVID-19.

The goal of our research study was to compare the effectiveness of several interventions in improving uptake of the COVID-19 vaccine in high-risk individuals. More specifically, the study aimed to identify the intervention channel and messaging approach that would result in the highest number of high-risk patients obtaining a COVID-19 vaccine during the study period. The study group consisted of patients at a network of 19 urban outpatient medical centers who were considered high-risk either due to age (>60 years old) and/or have a documented medical history of hypertension or diabetes. A secondary aim was to quantify self-reported COVID-19 vaccine status in these high-risk groups.

## 2. Materials and Methods

### 2.1. Patient Population and Sample Size

The patients enrolled in this study were drawn entirely from the patient population at Penda Health—a network of affordable, private outpatient medical centers in Nairobi, Kajiado, and Kiambu counties in Kenya. All facilities use a cloud-based electronic health record (Easy Clinic, Kolkotta, India). Our research team used our existing patient database to identify unique individuals who had previously been treated at a Penda Medical Center who were at least 60 years old and/or had hypertension or diabetes documented as a chronic illness in their records. In accordance with internal clinical practice guidelines, clinicians document hypertension as a chronic illness in patients with consecutive blood pressure readings demonstrating systolic pressure greater than 139 mmHg or diastolic pressure greater than 89 mmHg or both. Hypertension is also documented in the record if the patient is already on treatment with anti-hypertensive agents. Diabetes is documented as a chronic illness in patients with glycated hemoglobin (HbA1c) greater than 6.5% or random blood sugar levels greater than 11.1 mMol/L. Diabetes is also documented in patients on active treatment for diabetes mellitus with metformin, insulin, or other agents. We excluded individuals who had received any of the COVID-19 vaccines at any of our facilities.

According to the latest national census data, the number of adults residing in Nairobi county aged 60 and above is 102,106 [18]. Although some research exists on the prevalence of chronic illnesses in Kenya, there is no conclusive and specific data available on the total population size of Nairobi residents with a history of hypertension and/or diabetes. As such, we estimate that the total population relevant for this study is at least 65,143 given the current COVID-19 vaccination rates in Kenya (36.2% vaccinated adults). 

Based on previous literature demonstrating a maximum effect size of approximately 5% using SMS-based messaging [19], we estimated that from a sample of 100 unvaccinated individuals, we would see approximately five people choose to become vaccinated within the study period. Based on local vaccination rates of 36.2%, we anticipated needing about 276 people in each group to adequately assess our six intervention strategies as well as a control group. Given the known limitation of reaching individuals by phone (typically about 50%), we doubled the group size. To account for possible smaller effect sizes than anticipated as well as higher vaccination rates in high-risk sub-groups compared to the general population, we again increased the sample size to 1216 individuals per group (six intervention groups plus one control group), or an overall size of 8514 patients to conclusively demonstrate the effect of the interventions.

The total sample size from Penda Health patient health records was 8514 patients with a gender split of 57% females and 43% males. A stratified random sampling approach was used to generate the treatment groups with each patient belonging to one group only. More specifically, we stratified the participants in different treatment groups based on two features—gender and high-risk group—to ensure that each treatment group was representative of the sample. We used a Python (Pandas) function to perform the calculations and return the stratified groups.

### 2.2. Intervention Channels

Two intervention channels were used for in the study: (i)Outgoing Phone Calls: Patients in this treatment group received a phone call from a healthcare provider with the intended message. The phone calls were made by registered clinical officers fully employed at Penda’s Telemedicine Call Center. Calls were made using global call-center software (3CX, UK), and therefore appeared on patient phones as one consistent phone number that is known to be Penda’s contact number. If a patient could not be reached on the first attempt, health care providers would make at least one additional attempt later in the day. In the event of the patient calling back, the call would be routed to one of our healthcare providers to deliver the intended message.(ii)Bulk SMS Outreach: Patients in this treatment group received an SMS with the intervention message. SMSs were queued and loaded using a bulk messaging software (Africa’s Talking, Nairobi, Kenya) and were delivered on the same day.


### 2.3. Messaging Approaches

Three different messaging approaches were tested across both intervention channels. These messaging approaches leveraged on various behavioral mindsets of the study groups, and were designed through a collaboration between Penda Health and Dalberg Design, and informed by prior research on methods to mitigate vaccine hesitancy as well as local human-centered design testing approaches (unpublished).
(i)Gain Messaging: The focus of this approach was on highlighting a desirable or positive outcome of being vaccinated (e.g., spending time safely with family).(ii)Loss-Messaging: The focus of this approach was on emphasizing the undesirable or negative outcome of staying unvaccinated (e.g., the greater risk of negative health outcomes if you contract COVID-19 before having received the vaccine).(iii)Social proofing/norms: The focus in this approach was on relatability to actions by peers (e.g., many of your peers have already received the vaccine).


### 2.4. Treatment Groups

Based on the two intervention channels and three messaging approaches, six different treatment groups were created. A control group was also defined; the control group did not receive any intervention at all. Details of each treatment group and the control group are summarized in Table 1. The distribution of study population by risk category for each treatment group is presented in Figure 1.

All intervention treatments were implemented at the same time. Although the phone call intervention took much longer to complete compared to the SMS intervention, results were measured exactly one month after the intervention date. This ensured that each study participant was allowed the same amount of time to respond to the call to action.

## 3. Results

### 3.1. Reach via Phone Calls

Overall, 1716 (47%) participants in the phone call channel received intervention. This proportion of successful calls was remarkably similar across all three treatment groups with the lowest group at 559 (46%) participants and the highest group at 584 (48%) participants. This telephone response rate is similar to many other phone-based interventions we have conducted at Penda Medical Centers over several years.

For the remaining participants, 1240 (34%) did not answer any of the three attempted calls. No other follow-up was made for this group. The other 692 (19%) participants were completely unreachable, which means the call was automatically disconnected or the phone number was out of service. Unreachable contacts arise due to multiple SIM cards in use by a patient, deactivation of phone numbers, or incorrect mobile numbers captured during patient registration. Figure 2 summarizes reach for treatment groups in the phone calls channel.

### 3.2. Vaccination Status

For the successful phone calls, 1424 (83%) participants self-reported that they had already received the COVID-19 vaccine. A total of 1267 (89%) of the vaccinated participants had completed the vaccine doses while 157 (11%) had been partially vaccinated. The overall group of unvaccinated high-risk individuals eligible for our call to action was therefore considerably lower than we anticipated based on national statistics. Figure 3 summarizes the vaccination status reported by participants in the phone calls channel.

### 3.3. SMS Intervention Channel

For the treatment groups under the SMS channel, 44 patients (equivalent to 1% of total patients contacted) requested a call back for additional information. There was no significant difference in response rate according to the messaging approach. Among the 44 patients requesting a call back, the response rate for successful calls was 50%. Details of participants in the SMS channel who requested for a call back are presented in Table 2.

### 3.4. Overall Response to Call to Action

Out of the entire study group, four patients from the treatment group visited one of our outpatient medical centers for a COVID-19 vaccination within one month of the intervention. The table below summarizes conversion based on intervention channel and messaging approach. A summary of these results are presented in Table 3.

## 4. Discussion

This study provides several important findings for policymakers and clinicians. First, although the overall rate of vaccinated individuals in Kenya is low—only 36% as of October 2022—our study shows that higher-risk patients in urban areas such as those older than 60 or those with chronic illnesses have a remarkably high vaccination rate of 89%. Although we were delighted to find such a high vaccination rate among this high-risk group, we are also aware that urban areas, particularly within Nairobi county, have the highest vaccination rates in Kenya [14]. There may also be some selection bias in our sample of older and chronically ill individuals who have previously sought healthcare at a private medical facility that could have enriched our sample for particularly health-conscious individuals.

Second, among the unvaccinated individuals in our study, only four patients accessed the COVID vaccines within one month of our intervention, and most of those occurred in the Phone Calls—Loss-Messaging group. Although the sample size is relatively small, there was a clear trend toward loss-messaging being the most effective intervention. In contrast to the gain messaging recommended in the North American context, our findings align with survey-based research conducted in Kenya that showed that discussion of health outcomes of COVID in unvaccinated individuals (loss-messaging) was the most effective method by far—nearly three times more effective—as compared to social norming and economic messaging [17]. Although this topic clearly requires more study, our findings support other research that messaging approaches must be tailored to local context as much as possible. For local clinicians, our research suggests that brief, targeted discussions with unvaccinated patients about the health risks associated with COVID will be most effective in optimizing vaccine uptake.

Our finding that phone calls are much more effective, though more costly, as compared to bulk SMS messaging, also has important implications for policy makers. It is intuitive that a phone call is more personalized and comes across to the patient as a more direct recommendation from their health care provider as opposed to an SMS message. That being said, policy makers need to carefully weigh the costs of using phone calls as opposed to the extremely affordable bulk SMS approach.

We modeled the cost-effectiveness of these approaches by reviewing our internal operational costs and the efficacy of our interventions in this study. We estimate that each phone call from a registered clinical officer costs approximately 26 KSH (0.22 USD) at scale. If we were to start with a known unvaccinated group, we estimate that the cost of phone call outreach would be 2314 KSH (19.41 USD) per conversion, and 174 KSH (1.46 USD) per conversion using bulk SMS. This is a very important finding for policymakers and health system leaders since it demonstrates real-world evidence of an extraordinarily low-cost approach to increasing vaccination uptake in high-risk groups.

An important policy implication is that such outreach will be far more effective than the intervention described here by starting with a known unvaccinated group. This could be achieved at the national level using data from Kenya’s country-wide Chanjo system to target only high-risk unvaccinated individuals. According to a recent modeling study by the Kenya Medical Research Institute (KEMRI), the most cost-effective COVID-19 vaccination strategy for the country is to focus on the most at-risk groups for vaccination [20]. Our study provides real-world data to support a national approach to increasing COVID-19 vaccination rates among high-risk individuals. Furthermore, it suggests that using loss-messaging delivered by a trusted healthcare provider over a phone call will be the most effective, while an SMS using the same messaging approach will be the most cost-effective.

Our study had several limitations. First, the study would have been much more effective if we had started out with only unvaccinated high-risk adults. Due to limited data on vaccination records, this was not possible. Second, since participants could choose to be vaccinated outside of our network of medical centers even after the intervention, the number of true conversions is not completely certain. This could have led to an underestimation of the impact of the intervention. Third, our target groups were selected from a subset of the population who had received outpatient healthcare with us in the last five years. This group may have been positively selected for proactive health-seeking behaviors so the true COVID-19 vaccination rate among high-risk individuals across Kenya may be lower, particularly in rural areas. Finally, our data on vaccination status relied on self-report, and therefore may overestimate vaccination rate due to participants’ desire to respond in accordance with known public health recommendations.

In conclusion, our study demonstrates that COVID-19 vaccination rates in high-risk patient populations in urban Kenya are relatively high at 89%. This is a welcome finding for public health experts and policy makers in Kenya, though this finding needs to be replicated in rural populations throughout Kenya. Furthermore, we found that a loss-messaging approach was the most effective messaging strategy for this population. This result requires further validation due to the relatively low effect size of the intervention; however, the consistency of the greater impact of the loss-messaging approach across SMS and telephone outreach channels further supports our result. Finally, our results demonstrate that although a phone-call-based strategy is more effective than an SMS-based approach, a bulk-SMS channel is the most cost-effective method of improving vaccination rates in unvaccinated high-risk individuals in Kenya.

## Figures and Tables

**Figure 1 vaccines-11-00209-f001:**
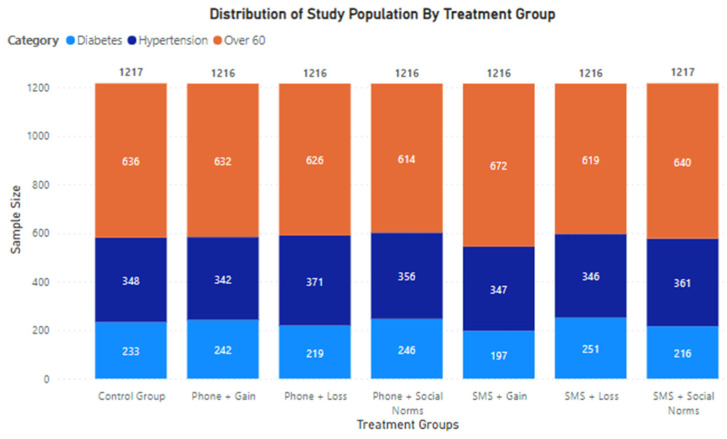
Distribution of treatment groups based on risk category.

**Figure 2 vaccines-11-00209-f002:**
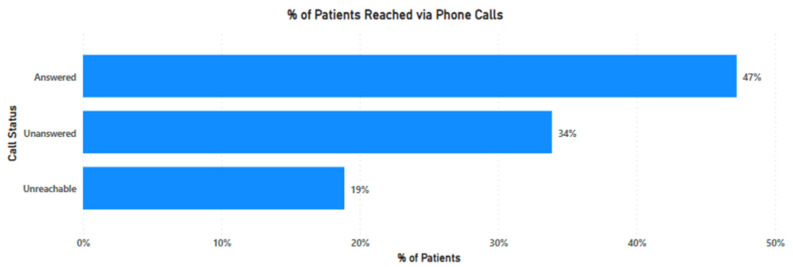
Reach for treatment groups in phone calls channel.

**Figure 3 vaccines-11-00209-f003:**
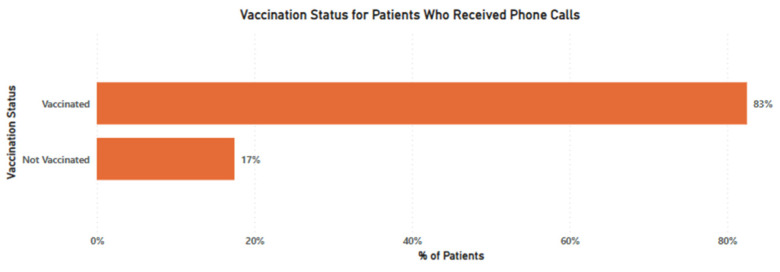
Self-reported vaccination status for participants reached through phone calls.

**Table 1 vaccines-11-00209-t001:** Summary of intervention provided by each treatment group.

Treatment Group	Intervention Channel	Messaging Approach	Script
Treatment Group 1	Phone Call	Gain Messaging	The Coronavirus is still present and active. Vaccination not only reduces the risk of complications from the virus but also gives you the confidence to visit family, friends and loved ones without anxiety or fear.
Treatment Group 2	Phone Call	Loss Messaging	The Coronavirus is still present and active. Studies show that those with underlying health conditions are more likely to suffer from severe infections, hospitalisations, and other serious health complications in the event that they contract COVID-19.
Treatment Group 3	Phone Call	Social Norms	As of today, over 350 patients in our network of medical centers with chronic conditions such as hypertension, diabetes, kidney disease, etc. have been vaccinated against COVID-19. Have you received the vaccine? OR As of today, over 830 patients in our network of medical centers above the age of 60 have been vaccinated against COVID-19. Have you received the vaccine?
Treatment Group 3	SMS	Gain Messaging	COVID-19 vaccination protects your health and gives you the confidence to visit family, friends and loved ones without anxiety or fear
Treatment Group 5	SMS	Loss Messaging	Studies show that underlying health conditions are likely to trigger infections and health complications as a result of COVID-19 infections
Treatment Group 6	SMS	Social Norms	Over 350 patients in our network of medical centers with chronic conditions such as hypertension, diabetes, kidney disease, etc. have been vaccinated against COVID-19 OR Message: Over 830 patients in our network of medical centers above the age of 60 have been vaccinated against COVID-19

**Table 2 vaccines-11-00209-t002:** Distribution of participants in the SMS group that requested additional information.

Messaging Approach	Total Unique Requests for Additional Information	As a % of Total SMS Sent
Gain Messaging	14	1.15%
Loss-Messaging	13	1.07%
Social Norms	17	1.40%
Total	44	1.21%

**Table 3 vaccines-11-00209-t003:** Summary results for participants who responded to the call to action.

Intervention Channel	Messaging Approach	Total Sample Size	Sample Reached	Total Eligible Unvaccinated Patients	Converted Patients Within 1 Month	Conversion Rate
Phone Calls	Gain Messaging	1216	583	84	0	0%
	Loss-Messaging	1216	555	110	3	2.37%
	Social Proofing	1216	585	73	0	0%
SMS	Gain Messaging	1216	n/a	n/a	0	0%
	Loss-Messaging	1216	n/a	n/a	1	0.08%
	Social Proofing	1217	n/a	n/a	0	0%
Control Group	n/a	1217	n/a	n/a	0	0%

## Data Availability

Data can be requested from the corresponding author.

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
