# Peer review of "A Comparison of Strategies to Improve Uptake of COVID-19 Vaccine among High-Risk Adults in Nairobi, Kenya in 2022"

_vaccines, 2023, doi:10.3390/vaccines11020209_

Round 1

Reviewer 1 Report

Dear Authors,

The researchers stated Covid-19 vaccine uptake in Kenya is still low compared to other countries especially in Europe and North America. In most parts of the country, a large percentage of the Kenyan population remains unvaccinated. As of October 2022, the Ministry of Health (Kenya) estimates that only 36.2% of the adult population had been fully vaccinated. Methods: We conducted an experimental study in April 2022 targeting unvaccinated adults who had a history of hypertension and/or diabetes and those in the 60+ age group. We tested various messaging approaches using two different intervention channels. Results: Although the overall rate of vaccinated individuals according to national records is low, responses from the study group collected through phone call conversations show that higher risk adults such as those older than 60 or those with chronic illnesses have a remarkably high vaccination rate of 89%. After the study, 4 participants received a covid-19 vaccine within 1 month of the intervention. These 4 participants all received a loss messaging intervention approach during the study. Conclusion: This study supports a national approach to increasing Covid-19 vaccination rates using loss messaging directed at unvaccinated, high-risk individuals.

By investigating the research paper, the current research is not very new as we have past works related to the field; nevertheless, the context of lectures is under-explored. The paper can offer insights pertaining to such context. Literature review is adequate. However, further recent literature could strengthen the current research. Methodology is clear; precise statistical analyses have been carried out with the research objectives. Findings are fairly presented and the analysis is presented in a good manner and presenting new ideas. The research paper lacks managerial implications. It will be better to have several separate sections that describe implications for academics and practitioners. It needs some improvements in order to meet the quality standards of the respected journal. Also, references should be checked and the et al’s should be added in References 2, 4, 6, 10, 14 & 17.

Author Response

We thank the reviewers for the time and effort they have invested in helping us to improve our manuscript.

We thank Reviewer 1 for the constructive comments regarding the manuscript.  In our revision, we have expanded our literature review in the introduction and background section in order to strengthen the contextual grounding of our work, and we agree this has improved the quality of the manuscript.  We have also addressed the reviewer’s concerns about the citations to ensure the authors names are enumerated in accordance with the expected citation style.  Reviewer 1 made an excellent observation that the manuscript would be strengthened by clearly delineating managerial/policy implications from clinical practice implications.  We agree, and we have amended the manuscript’s discussion section to incorporate this helpful framework.

Reviewer 2 Report

This article engages in a broad discussion to test the effectiveness of various interventions in increasing the uptake of the Covid-19 vaccine in high-risk individuals. More specifically, this study aims to identify the channel of intervention and message delivery that will result in the highest response rate in terms of vaccination. To test several hypotheses about the medium of intervention and message delivery, understood as a series of decisions in high-risk individuals, as predictors of uptake of the Covid-19 vaccine, the author's used survey data.

I have great concern about articles that make me uncomfortable while reading them. The approach you present can be classified as a «high-risk individual blame» approach, which has been thoroughly discussed in the introduction and discussion. While one does not question the freedom of researchers to adopt a theoretical point of view they deem more appropriate, the failure to acknowledge the controversial character of the theoretical framework they have chosen is inconsistent with the general practice of science. I would encourage authors to consider these preliminary revisions and discussions more carefully and to include in their articles a meeting of the responses that have been put forward to explanatory theories that focus on high-risk individuals and address these criticisms from their point of view.

The methods section also needs to be revised to include some considerations on measuring high-risk individuals. The scales used by the authors do not necessarily measure high-risk individuals who precede the uptake of the Covid-19 vaccine but are a consequence of living in vulnerability. This complex causality should be discussed or at least acknowledged. This has immediate implications for the findings and discussion. The sampling is also not explicit; please distinguish between the population and the sample.

When presenting the sociodemographic characteristics of the sample, the authors included individuals at high risk, but it is not clear how to measure them. Some references, such as thresholds, would be welcome to help readers unfamiliar with the state understand what values ​​represent.

Author Response

We thank the reviewers for the time and effort they have invested in helping us to improve our manuscript.

We thank Reviewer 2 for the candid comments regarding the theoretical framework used to approach high-risk individuals.  We appreciate the reviewer for pointing out how our approach could come across as controversial given the vulnerable nature of the high-risk group we studied, and that we must explicitly comment on this dynamic to ensure there are no implications of blame on the individuals we studied.  We hope the changes made in the introduction and discussion section of our manuscript adequately address the concerns raised, and we appreciate the feedback.  

We have also added more detailed criteria regarding how we defined high-risk individuals and we have provided clarification on our sampling approach.

Reviewer 3 Report

this study is interesting. The abstract is enough. The materials and methods well described. Also results. The discussion and conclusion are enough. The references could be implemented.

Author Response

We thank Reviewer 3 for the time and effort invested in helping us to improve our manuscript. We have addressed Reviewer 3’s suggestions for improving the references in our updated manuscript.

Round 2

Reviewer 2 Report

The author has made revisions according to the directions of my review, so I consider this worthy of acceptance.

Author Response

Thank you.